# Unique Structural Features Relate to Evolutionary Adaptation of Cytochrome P450 in the Abyssal Zone

**DOI:** 10.3390/ijms26125689

**Published:** 2025-06-13

**Authors:** Tatiana Y. Hargrove, David C. Lamb, Zdzislaw Wawrzak, George Minasov, Jared V. Goldstone, Steven L. Kelly, John J. Stegeman, Galina I. Lepesheva

**Affiliations:** 1Department of Biochemistry, Vanderbilt University School of Medicine, Nashville, TN 37232, USA; tatiana.y.hargrove@vanderbilt.edu; 2Faculty of Medicine, Health and Life Science, Swansea University, Swansea SA2 8PP, UK; d.c.lamb@swansea.ac.uk (D.C.L.); s.l.kelly@swansea.ac.uk (S.L.K.); 3Northwestern Synchrotron Research Center, Northwestern University, Argonne, IL 60439, USA; zw@ls-cat.org; 4Department of Microbiology-Immunology, Northwestern University Feinberg School of Medicine, Chicago, IL 60611, USA; g-minasov@northwestern.edu; 5Department of Biology, Woods Hole Oceanographic Institution, Woods Hole, MA 02543, USA; jgoldstone@whoi.edu (J.V.G.); jstegeman@whoi.edu (J.J.S.); 6Center for Structural Biology, Vanderbilt University, Nashville, TN 37232, USA

**Keywords:** cytochrome P450, sterol 14α-demethylase (CYP51), crystal structure, structure–function, conformational flexibility, biological membranes, evolutionary adaptation

## Abstract

Cytochromes P450 (CYPs) form one of the largest enzyme superfamilies, with similar structural folds yet biological functions varying from synthesis of physiologically essential compounds to metabolism of myriad xenobiotics. Sterol 14α-demethylases (CYP51s) represent a very special P450 family, regarded as a possible evolutionary progenitor for all currently existing P450s. In metazoans CYP51 is critical for the biosynthesis of sterols including cholesterol. Here we determined the crystal structures of ligand-free CYP51s from the abyssal fish *Coryphaenoides armatus* and human-. Comparative sequence–structure–function analysis revealed specific structural elements that imply elevated conformational flexibility, uncovering a molecular basis for faster catalytic rates, lower substrate selectivity, and intrinsic resistance to inhibition. In addition, the *C. armatus* structure displayed a large-scale repositioning of structural segments that, in vivo, are immersed in the endoplasmic reticulum membrane and border the substrate entrance (the FG arm, >20 Å, and the β4 hairpin, >15 Å). The structural distinction of *C. armatus* CYP51, which is the first structurally characterized deep sea P450, suggests stronger involvement of the membrane environment in regulation of the enzyme function. We interpret this as a co-adaptation of the membrane protein structure with membrane lipid composition during evolutionary incursion to life in the deep sea.

## 1. Introduction

Cytochromes P450 (CYP, P450) are ubiquitous enzymes found throughout nature with various biological functions. In metazoans P450s participate in the synthesis and metabolism of sterols, steroid hormones, prostaglandins, vitamins A and D and bile acids, as well as catalyze the monooxygenation of various secondary metabolites and xenobiotics such as drugs, pollutants and carcinogens [1]. Sterol 14α-demethylases (CYP51s) represent a very special P450 family, which is regarded as the possible evolutionary progenitor for all currently existing P450s. CYP51 enzymes are found in all kingdoms of life, catalyzing the same complex, three-step stereo- and regio-specific reaction that is essential for the biosynthesis of sterols, such as cholesterol in humans and other vertebrates, and ergosterol in pathogenic fungi and protozoa, whose CYP51 orthologs are targets for azole drugs [2].

To date, crystal structures have been determined for CYP51s from 13 organisms, including -human and 12 unicellular species: three fungal pathogens (*Candida* and *Aspergillus* spp.), five protozoan pathogens (*Trypanosoma* spp., *Leishmania*, *Naegleria*, *Acanthamoeba*) and three bacteria *Mycobacterium tuberculosis*, *Methylococcus capsulatus*, *Mycobacterium marinum* (e.g., [3,4,5,6,7,8,9,10,11,12,13]). The crystal structures are gradually uncovering the molecular basis for this enzyme’s strict functional conservation preserved across hundreds of millions of years of evolutionary diversification. Thus, we have found that despite very low sequence identity across phylogeny (often less than 30%), at the three-dimensional level CYP51s display high structural similarity and do not reveal large structural rearrangements when co-crystallized with various inhibitory ligands [2]. Binding of physiological substrates, however, causes a conserved conformational switch that involves closing of the active site entrance (bordered by helices A’, F’’ and the tip of the β4 hairpin) [14,15,16]. The precise molecular details of how the substrate enters the P450 active site from the endoplasmic reticulum membrane where the substrates are located, how the product is released back into the membrane environment, and what the roles of membranes in these processes are remain elusive.

Here we report the first crystal structure of a non-mammalian vertebrate CYP51, from the abyssal grenadier (*Coryphaenoides armatus*), a deep-sea fish common in most oceans of the world at depths between one and five kilometers. Sequence–structure–function comparison of *C. armatus* and human CYP51s with their microbial counterparts reveals specific molecular features with implications for (1) supporting the notion that in animals the CYP51 evolutionary advantage was to achieve faster catalysis, and (2) that this feature probably developed as an increase in molecular flexibility. Furthermore, the data suggests that not only protein folding [17] but also interactions with the highly specific pressure-resistant fish membranes afford the *C. armatus* CYP51 ortholog the ability to perform its catalytic function while withstanding the high hydrostatic pressure of 400–500 atm and nearly freezing temperatures of the deep sea. The findings also highlight limitations of relying on modelling techniques, such as AlphaFold [18] and high level homology modelling (e.g., Modeller) to understand P450 architecture because the unique structural properties revealed by X-ray crystallography were not seen when the proteins were modelled [17].

## 2. Results

### 2.1. Sequence Analysis

To compare the CYP51 sequences from different phyla, Clustal Omega alignment was performed, starting from the proline-rich region (following the N-terminal membrane anchor sequence, which is least conserved in CYP51 enzymes), P61 in the human ortholog. Although the N-terminal anchor in *C. armatus* CYP51 is one residue shorter, for simplicity, here and throughout the manuscript we are using the human CYP51 amino acid numbering. Statistical analyses of the catalytic domains show 80% identity between *C. armatus* and human CYP51s (90% similarity). The identity of *C. armatus* CYP51 to other fish species ranges from 89% (*Gadus morhua*, Atlantic cod, XP_030203989_1, 96% similarity) to 78% (*Danio rerio*, zebrafish, NP_001001730.2, 89% similarity). The identities to the plant and eukaryotic microbial orthologs drop to 38–32% (e.g., *Arabidopsis thaliana*, NP_172633.1, 38%; *Naegleria fowleri*, KAF0972476.1, 36%; *Candida albicans*, XP_716761.1, 35%; *Ajellomyces capsulatus*, EER42982.1, 32%). The lowest identity the *C. armatus* CYP51 ortholog has is to the CYP51s from Trypanosomatidae (*Trypanosoma cruzi*, AY856083.1, 30%) and bacteria (*Mycobacterium ulcerans*, WP_011738801.1, 29%; *Sandaracinus*-NAT131, MAT24795.1,27%; *Candidatus Microthrix parvicella*, WP_020378057.1, 24%). Multiple sequence alignment of the full-length CYP51 proteins from *C. armatus*, human, and other fish species can be seen in Appendix A.

### 2.2. Substrate Binding Profile and Catalytic Parameters

We found that enzymatically, both in terms of substrate requirements and catalytic properties, *C*. *armatus* CYP51 reveals close resemblance to the human ortholog. Unlike plant and most microbial sterol 14α-demethylases, *C*. *armatus* CYP51 binds all the tested CYP51 substrates (including mammalian lanosterol and 24,25-dihydrolanosterol, fungal 24-methylene-24,25-dihydrolanosterol and plant obtusifoliol) with similar binding affinities and catalyzes their 14α-demethylation with similar catalytic efficiency (Figure 1 and Table 1).

Moreover, similar to what we previously observed under the same conditions with human CYP51 [3], the amplitudes of the type I spectral response of *C. armatus* CYP51 to the binding of either of the sterol substrates (due to the blue shift in the Soret band maximum) do not exceed 30% of the maximal low-to-high spin state transition in the P450 heme iron. This suggests a sufficiently large void volume of the active site so that the iron-bound water molecule does not necessarily have to be completely expelled from the iron coordination sphere for the substrate to be properly accommodated. The values of spectrally determined apparent dissociation constants (K_d_) are within 0.6–1.0 µM, with similar binding efficiencies (ΔA_max_/K_d_). The *k*_cat_ ranges of *C. armatus* (48–62 nmol/nmol/min) and human CYP51s (45–64 nmol/nmol/min, determined at the same reaction conditions [3]) are also comparable and are amongst the fastest reported for a CYP51 enzyme. This supports the hypothesis that the evolution of sterol biosynthesis in vertebrates might have been directed toward faster sterol flow, even if at the expense of substrate selectivity.

### 2.3. Intrinsic Resistance to Inhibition

Azole antifungals whose primary mode of action is to inhibit microbial CYP51s have been shown to be environmental pollutants, in freshwater and marine ecosystems [19,20]. Due to their widespread occurrence in the environment, there are major concerns over their detrimental effects on aquatic organisms. *C. armatus* CYP51, however, again similar to the human ortholog, displayed low susceptibility to inhibition, although its resistance to imidazole-based ketoconazole and our experimental compound VFV [(*R*)-*N*-(1-(3,4′-difluorobiphenyl-4-yl)-2-(1*H*-imidazol-1-yl)ethyl)-4-(5-phenyl-1,3,4-oxadiazol-2-yl)benzamide]^2,3,8^ was relatively weaker than that of the human enzyme. Thus, at a 50-fold molar excess of ketoconazole in a reconstituted 1 h reaction (the conditions producing a complete inhibition of microbial CYP51s even at a 2-fold molar excess of inhibitor [5,8,21]), *C. armatus* CYP51 converted 37% of lanosterol, while human CYP51 under these conditions converts 100% [3]. The inhibition of *C. armatus* CYP51 with VFV was 93%, while the inhibition of human CYP51 was 64% [3]. The triazole-based antifungal drugs fluconazole, voriconazole, itraconazole, isavuconazole and posaconazole, as well as the tetrazoles oteseconazole (VT-1161) and VT-1598, did not have any inhibitory effect on the *C. armatus* CYP51 activity (Figure 2). Based on our previous crystallographic studies [5], the differences might be related to the strength of a coordination bond between the P450 heme iron and the azole nitrogen, which is more polar in the imidazole ring than the nitrogen of triazoles and tetrazoles.

Spectral responses of *C. armatus* CYP51 to VFV, ketoconazole, itraconazole and voriconazole are shown in Appendix A, with the apparent dissociation constants (K_d_) being 0.02, 0.01, 0.62 and 0.61 µM. The corresponding values for human CYP51 are 0.09, 0.07, 1.29 and 2.53 µM, respectively [3]. This indicates that, like human CYP51, the *C. armatus* ortholog binds the azole drugs, but during the catalytic cycle they are being easily displaced in the active site by the sterol substrate.

### 2.4. Structural Characterization

To better understand sequence–structure–function interrelations within the CYP51 family, we determined the ligand-free crystal structures of *C. armatus* and human CYP51s. The *C. armatus* CYP51 enzyme was crystallized in the P12_1_1 space group; the structure was refined to 2.9 Å, with the R_work_/R_free_ = 0.232/0.248 and the average B-factor 98 Å^2^ (PDB code 9BAT). Human CYP51 was crystallized in the P2_1_2_1_2_1_ space group; the structure was refined to 2.7 Å, with the R_work_/R_free_ = 0.230/0.246 and the average B-factor 51 Å^2^ (PDB code 8SBI) (Table 2). In both P450 structures, the asymmetric unit consisted of two protein molecules. The average RMSD between the Cα atoms in the two molecules of *C. armatus* CYP51 was 0.43 Å, and the average RMSD between the Cα atoms in the two molecules of human CYP51 was 0.41 Å. The average RMSD between the Cα atoms in the *C. armatus* and human structures was 0.97 Å.

The structural similarities that we find distinguishing for both ligand-free vertebrate CYP51s include the “broken I-helix” (residues 311–313), and the “heme bulge” segment (residues 441–450) shifted away from the P450 heme (Figure 3). The tendency of the middle portion of the I helix in human CYP51 to adopt a loop-like conformation, even when the enzyme is in complexes with the relatively weak inhibitors (for example VFV and ketoconazole) but not with the substrate [16], has been noticed previously [3]. We assumed that this might be caused by the long sequence of polar residues (-T_315_SSTTS_320_-), which disrupt normal alpha-helical hydrogen bond pattern, destabilizing the helix and raising the active site flexibility, which in turn makes the protein more resistant to inhibition. We also found that a single amino acid mutation in human CYP51 (T318I, introduced to make the P450 sequence more microbial-like) [22] substantially increased its susceptibility to inhibition. Since the -TSSTTS- sequence is conserved across all known CYP51s from vertebrates (while in microbial CYP51 sequences it is always interrupted by one or more hydrophobic residues), we supposed that this feature of the I helix might be a general feature for vertebrate sterol 14α-demethylases. The high drug resistance of *C. armatus* CYP51 enzyme, revealed in our reconstituted reactions in vitro, and the loop-like area, found in the catalytic portion of its I helix in the ligand-free *C. armatus* CYP51 structure, strongly support this hypothesis.

The shifted heme bulge segment in the ligand-free *C. armatus* and human CYP51 structures is accompanied by the loss of the conserved heme-supporting salt bridge between the imidazole nitrogen of H447 and the carboxylate oxygen of the porphyrin ring D propionate (Figure 3B). The lack of this salt bridge and a loop-like shape (disrupted H-bonds) of the heme bulge are also indicative of a relatively higher conformational flexibility, although we currently cannot find a clear (local) sequence-based explanation for this observation because this region is highly conserved in CYP51 enzymes. Notably, this shift in the heme bulge segment as well as the disordered fragment in the I helix are not seen in the human CYP51 model from the AlphaFold Protein Structure Database (Appendix A, AF-Q16850-F1). We also noted that of the three conserved CYP51 supplemental cavities (voids) that were missing in the *C. armatus* CYP51 model [17], one void (171 Å^3^) is actually present in the 9BAT structure, again pointing to novelties not captured by structure-based modeling predictions.

Except for these two peculiarities, the heme environment in the two CYP51 structures remains the same. On the proximal side of the protein molecule, the heme iron is coordinated to C449 (2.2 Å). On the distal side (substrate binding pocket), the ring D propionate interacts with Y145 (helix B’) and K156 (helix C), and the ring A propionate interacts with the guanidinium group of R382 (β1–4 strand) with the side chain of Y131 (B’C loop), which is known to acquire various conformations in CYP51 enzymes [4,6], being positioned 3.7 Å from the heme (Figure 4).

There are, however, two striking differences between the fish and human CYP51 structures, and those are in the location of the β4 hairpin (residues 481–494) and, even more so, in the location of the FG arm (residues 232–258) (Figure 5). The middle portion of the FG arm in the fish enzyme lacks the characteristic F’’ helix. Instead, this region of the molecule adopts a loop-like conformation and is immersed deep inside the active site pocket, approaching helix I (SRS4 [23]), strand β1–4 (SRS5), and “clashing” with the β4 hairpin (SRS6) in the superimposed human CYP51 structure. In turn, the *C. armatus* CYP51 β4 hairpin is moved outward, away from the active site, protruding above the surface of the protein globule. The distance between the *C. armatus* P242 and the corresponding proline (ending the F” helix in the human structure) is 24 Å, and the tips of their β4 hairpins (I488) are 16 Å apart (Figure 5, inset).

The arrangements of these two secondary structural elements found in the *C. armatus* CYP51 structure are unique, not seen in other CYP51 orthologs to-date. The FG arm is thought to be generally flexible in cytochromes P450 [24,25], and was predicted to open wider in CYP51s for their substrates (as well as other ligands) to enter the active site [8]. We have now observed its large-scale inward movement, which is accompanied by the outward movement of the β4 hairpin. To exclude that these relocations in the *C. armatus* CYP51 structure may have happened due to some drastic crystal-packing events and protein denaturation (P450 to P420 conversion), crystals were dissolved, and the absorption spectra of the dissolved crystals (both absolute and difference CO-binding) were taken, showing that the *C. armatus* protein remained in its active P450 form throughout all the undertaken experiments (Appendix A), thereby confirming that the conformational dynamics of such an extent -is real and must be allowed in this enzyme in nature.

Further comparative analysis of human and *C. armatus* CYP51s reveals that in the primary sequences of their FG arm there are three amino acid differences, 228A/C, 237A/E and 243G/S (human/*C. armatus*, respectively) (Figure 5), with the fish CYP51 residues (C228 and E237) having lower propensities to form helical structures than the corresponding alanines in human CYP51. It is noteworthy that the same three substitutions are present in CYP51 sequences from other *Gadiformes* fish species that also live in deep or cold waters (e.g., Arctic cod (up to 1000 m, frequently under ice), Patagonian moray cod (up to 900 m, ~4 °C), Alaska pollock (up to 1300 m, 2–4 °C), Benguela hake (up to 900 m) and burbot (below 300 m) in Appendix A while shallow water fish CYP51s often have the same residues as the human ortholog. In turn, the tip of the β4 hairpin in *C. armatus* CYP51 is more hydrophobic, I482 (conserved amongst fish) instead of human V, F484 (conserved in deep-water fish) instead of Y (human and shallow-water fish) and N492 (mostly H in fish CYP51s) instead of E in the human sequence (Figure 5 and Appendix A). Higher residue hydrophobicity here can be indicative of stronger interactions with the hydrophobic tales of fatty acids in the fish membranes.

## 3. Discussion

Evolutionary colonization of the deep sea required molecular adaptations for enzyme function under high pressure and low temperature. There is a long history of efforts to identify them and understand their functional implications (e.g., [26,27]). Few studies have addressed adaptation of deep-sea proteins by direct determination of structure, and to our knowledge no structures have been determined for any deep-sea membrane-bound enzymes. This first non-mammalian vertebrate CYP51 structure offers new molecular insights into CYP51 catalysis and identifies structural modifications likely contributing to its function at depth. Comparison of the structures of *C. armatus* and human CYP51s suggests that the P450 FG arm and β4 hairpin are responsible for substrate capture and their flexibility must be particularly important in the deep-sea environment. Because in vivo in eukaryotic CYP51s these structural segments are immersed in the ER membrane (Figure 6), we surmise that the reason for the unique features of *C. armatus* CYP51 must be related to the differences in the composition of the fish membranes.

The *C. armatus* species are found at depths of up to 5500 m, where the pressure reaches 550 atm, and the water temperature is consistently 2–4 °C. It is known that high hydrostatic pressure and low temperature exert similar ordering effects on biological membranes, reducing their fluidity by increasing the packing of fatty acyl chains [28]. To compensate for these ordering effects, the biochemical composition of the membrane lipids in organisms that have evolved in extreme thermal or abyssal habitats is modified [29,30,31]. Their phospholipids have a higher proportion of unsaturated and polyunsaturated (animals) or branched-chain (microbes) fatty acids [32,33,34,35,36], which control the membrane thickness, fluidity, and microviscosity, providing insulation for membrane-bound enzymes [37]. In their free form, highly unsaturated fatty acids (with ≥20 carbons and ≥3 double bonds) have a very low melting point and thus have a much greater tendency to remain fluid [38]. This is because the presence of double bonds introduces kinks in the fatty acid chains. Analysis of deep-sea fish phospholipid composition revealed that, at 4000 m, the percent of saturated fatty acids was reduced by about 1.5-fold in phosphatidylcholine and more than 3-fold in phosphatidylethanolamine in comparison to the shallow water species [32]. In the liver of *C. armatus,* the principal unsaturated fatty acids are docosahexaenoic, 22:6 (4,7,10,13,16,19), n-3, melting point −44 °C; eicosapentaenoic, 20:5 (5,8,11,14,17), n-3, melting point −54 °C; and arachidonic, 20:4 (5,8,11,14), n-6, melting point −49 °C [30]. It has also been found that high pressures and low temperatures enhance glycosylation of membrane proteins [39], which in turn improves their stability and activity [40,41,42,43].

There are also multiple reports on accumulation of specific protein-stabilizing solutes (osmolytes) such as trimethylamine oxide (TMAO), whose intracellular concentration increases in abyssal organisms, resisting pressure-induced structural perturbations such as protein unfolding and cell dehydration [44,45,46,47,48]. It was reported that the concentration of TMAO in teleost fishes increases with depth, going from 40 to 261 mmol/kg from 0 to 4850 m [49]. In our in vitro experiments, TMAO had a stabilizing effect on *C. armatus* CYP51, as registered by the CO-binding spectra (P450 recovery after overnight incubations in the TMAO sample being 90% vs. 22% in the sample without TMAO, Figure 7A), as well as in the time course reaction of lanosterol 14α-demethylation (Figure 7B). As TMAO does not cause any spectral response in the *C. armatus* CYP51 heme iron, we assume that it is not binding inside the P450 active site, and the observed stabilizing effect must be produced on the surface, perhaps also involving protein/lipid interface. Our attempts to co-crystallize *C. armatus* CYP51 with TMAO were unsuccessful.

In conclusion, it is known that the membranes in the deep-sea organisms remain functional under high pressure and low temperature due to their specific lipid composition, as well as stronger regulation of membrane organization and protein stabilization [50]. As a way of convergent adaptation to the extreme environment, deep-sea fish (and other organism) membranes are more ordered, and the interactions between the lipid components and membrane proteins are stronger [28,31,51,52,53]. The proteins evolve to rely upon the enhanced membrane order, becoming more membrane-dependent. As a result, in the absence of proper environmental constraints (at ambient pressure and temperature), the portions of the protein molecules that in situ are membrane-embedded display higher flexibility than their counterparts isolated from the membranes of surface-dwelling organisms that live at atmospheric pressure. The unusually large-scale conformational rearrangements, observed here for the first time, must be allowed in CYP51 enzymes (as seen in Appendix A, the *C. armatus* protein remains in the P450 form). Furthermore, it is probable that such large motions are catalytically required in vivo (e.g., promoting the membrane-guided substrate binding and the product release) and have not been captured by X-ray crystallography before, as it only reveals the snapshots of predominant conformational states with the lowest energy. Applications of cryo-EM techniques for these purposes can be more beneficial. To fold and function at their physiological conditions, piezophilic proteins must have conformational landscapes at high pressure similar to those of their non-piezophilic homologs at ambient pressure. Placing such a protein at ambient pressure displaces its native conformational equilibrium towards low-populated states, which may be transient in catalysis [54]. It would be logical to hypothesize that the conformation of CYP51 from *C. armatus* observed here does not represent the predominant conformation of the substrate-free enzyme at the pressure of habitation of the fish. Instead, it is a transient, low-populated conformation common to all CYP51s, which becomes more populated due to the placement of the high-pressure adapted enzyme at ambient conditions. Technology for crystallization under pressure is being considered.

Overall, the structural distinction of *C. armatus* CYP51 must be a result of co-adaptation of membrane protein structure with changes in membrane phospholipid composition during evolutionary incursion into life in the deep sea. Having uncovered this feature of CYP51 from an abyssal fish raises the question as to whether other P450s from deep-sea species may show similar structural features and how these features might vary with different degrees of evolutionary adaptation to high pressure.

## 4. Materials and Methods

### 4.1. Protein Expression and Purification

Recombinant human CYP51 and rat cytochrome P450 reductase (CPR) were expressed and purified as described previously [3,55]. The expression and purification of CYP51 from *C*. *armatus* was performed following the human CYP51 protocol. Briefly, the *C. armatus* CYP51 cDNA, subcloned into the pCW expression plasmid using the NdeI/HindIII restriction sites [17], was transformed into *E. coli* HMS174 (DE3) (Novagen) and cultured in Terrific Broth (TB) media supplemented with 100 mM potassium phosphate buffer (pH 7.2) containing 0.1 mg/mL ampicillin and 125 µL of trace elements salt solution at 26 °C. Following induction with 1 mM IPTG and 1 mM δ-aminolevulinic acid, the bacterial cells were harvested and pellet homogenized in 50 mM potassium phosphate buffer (pH 7.2) containing 100 mM NaCl, 0.1 mM EDTA, 10% glycerol (*v*/*v*) and 0.1% Triton X-100 (*v*/*v*), sonicated (Sonic Dismembrator model 500, Fisher Scientific, Hampton, NH, USA) and stirred at 4 °C for 1 h. The solubilized protein was separated from the insoluble material by centrifugation at 82,000× *g* for 40 min (Optima L-80 Ultracentrifuge, Beckman, Brea, CA, USA) and purified in two steps, including affinity chromatography on Ni^2+^-NTA agarose (eluted with a 30 to 200 mM imidazole gradient in 20 mM potassium phosphate (pH 7.2) containing 200 mM NaCl, 10% (*v*/*v*) glycerol, and 0.1 mM EDTA) followed by cation-exchange chromatography on CM Sepharose (eluted with 500 mM NaCl in 20 mM potassium phosphate containing 10% (*v*/*v*) glycerol and 0.1 mM EDTA). The protein with the spectrophotometric index 417/278 > 1.2 was aliquoted, frozen in liquid nitrogen and stored at −80 °C until use. The yield was about 500–600 nmol/liter culture. The purity was verified by SDS-PAGE.

### 4.2. UV-Visible Spectroscopy

Absorption spectra (270–700 nm) were recorded at 22 °C using a dual-beam Shimadzu UV-2600i spectrophotometer and UVProbe2.71 software. The P450 concentration was determined from the Soret band absorbance in the absolute spectrum, with the extinction coefficient 117 mM^−1^ cm^−1^ for the low-spin ferric form or 91 mM^−1^ cm^−1^ for the reduced (ferrous) carbon monoxide (CO) complex in the difference spectra [56]. The spin state of the P450 samples was estimated from the absolute spectra using the ratio (ΔA_393–470_/ΔA_417–470_), with values of 0.35 and 2.0 corresponding to 100% low- and 100% high-spin iron, respectively.

### 4.3. Spectral Titrations with Sterol Substrates and Azole-Based Ligands

Lanosterol, 24,25-dihydrolanosterol (the natural CYP51 substrates in vertebrates), obtusifoliol (plants), and 24-methylene-24,25-dihydrolanosterol (fungi) were added from the 0.5 mM stocks in 45% water solution of 2-Hydroxypropyl-β-cyclodextrin (HPCD) to the sample cuvette with ~2 µM P450 in the 50 mM potassium phosphate buffer (pH 7.4) containing 100 mM NaCl and 0.1 mM EDTA; the titration step was 1 µL (0.25 µM). Equal amounts of the HPCD solution were added to the reference cuvette to correct for the solvent-induced spectral perturbations, and the difference spectra were taken in the wavelength range of 350–500 nm. The apparent spectral dissociation constants of the enzyme-substrate complex (*K*_d_) were calculated in GraphPad Prism (version 6.05) software by fitting the data for the substrate-induced absorbance changes Δ(A_390_–A_423_) versus substrate concentration to a one site-total binding equation (binding-saturation). Titrations with azole-based compounds were carried out at 0.5 µM P450 concentration in 5 cm optical path length cuvettes. Aliquots of 0.1 mM compounds dissolved in dimethyl sulfoxide (DMSO) were added to the sample cuvette, with each titration step being 0.1 µM. At each step, the corresponding volume of DMSO was added to the reference cuvette. The apparent spectral dissociation constants (*K*_d_) were calculated in GraphPad Prism software, as described, for human CYP51 [22].

### 4.4. Reconstitution of Catalytic Activity, Kinetic Analysis and Inhibition

Activity assays were generally performed as previously described for human CYP51 [3] using the radiolabeled ([3-^3^H]) sterol substrates lanosterol, 24,25-dihydrolanosterol, 24-methylene-24,25-dihydrolanosterol, and obtusifoliol. Time-course experiments were carried out at 37 °C at 25 μM concentrations of sterol substrates, 0.25 μM P450, and 1 μM CPR. For steady-state kinetic analysis, the reactions were run for 1 min with the sterol concentrations 3.13, 6.25, 12.5, 18.75, 25, 31.25 and 37.5 µM. The samples were preincubated for 3 min at 37 °C in a shaking water bath, and the reaction was initiated by the addition of 100 μM NADPH and stopped by the extraction of sterols with ethyl acetate. The extracted sterols were dried, dissolved in 100 µL of methanol, and the products were analyzed by reversed-phase HPLC equipped with a *β*-RAM detector (INUS Systems) using a Nova-Pak 3.9 mm × 150 mm (4 μm) octadecylsilane (C18) HPLC column. The products were separated with an isocratic mobile phase composed of acetonitrile and methanol (4:1 (*v*/*v*)) at a flow rate of 0.75 mL/min [9]. Michaelis–Menten parameters were calculated using GraphPad Prism. The *k*_cat_ and *K*_m_ values for each reaction were determined by fitting the data to a Michaelis–Menten hyperbola, with the reaction rates (nmol product formed/nmol P450/min) being plotted versus total substrate concentration. Susceptibility of *C. armatus* CYP51 to inhibition with azole antifungal drugs and VFV was determined as a decrease in the substrate conversion in a 1 h reaction at the enzyme/inhibitor/substrate molar ratio of 1/50/50 and P450 concentration 0.5 µM [3].

### 4.5. X-Ray Crystallography

The crystals of ligand-free *C. armatus* CYP51 were obtained using the sitting drop vapor diffusion technique at 18 °C. The drops with the average P450 concentration of 200 µM, containing 20 µM n-tetradecyl-β-D-maltoside (Hampton Research, Aliso Viejo, CA, USA), were overlayed with the equal volume of well solution (0.2 M magnesium acetate and 25% (*w*/*v*) PEG 3350, pH 7.7). Crystals appeared overnight, were cryoprotected with mother liquor containing (*v*/*v*) 25% glycerol and flash-cooled in liquid nitrogen. The crystals of ligand-free human CYP51 (the D205A/H314A mutant [16]) were obtained by the hanging drop vapor diffusion technique at 23 °C. The drops with the average P450 concentration of 300 µM, containing 2.5 mM n-decanoyl sucrose (Hampton Research), were mixed with an equal volume of well solution (0.2 M calcium acetate hydrate and 20% (*w*/*v*) PEG 3350, pH 7.3). Crystals appeared within 4 days and were cryoprotected by soaking them in mother liquor with 40% (*v*/*v*) glycerol and flash-cooled in liquid nitrogen.

The data were collected at 100 K using synchrotron radiation at the European Synchrotron Radiation Facility, Grenoble, for the ID-23-2 beamline (*C. armatus* CYP51) and at the Advanced Photon Source, Argonne National Laboratory, for the 21-ID-F beamline (human CYP51). The diffraction images were indexed and integrated with autoProc [57] and scaled with Aimless [58]. The structures were determined by molecular replacement with PhaserMR in the CCP4 program suite [59], using 4UHI as the search model for 8SBI and 8SBI as the search model for 9BAT. The refinement and model building were performed with Refmac5 (CCP4) and Coot [60], respectively. The data collection and refinement statistics are shown in Table 2. The coordinates and structure factors were deposited in the Protein Data Bank. Structural comparisons were accomplished and RMSDs calculated in LSQkab (CCP4) using a secondary structure matching algorithm.

### 4.6. Spectral Characterization of the Dissolved C. Armatus CYP51 Crystals

After data collection the crystals were dissolved in phosphate buffered saline (PBS, Gibco, Billings, MT, USA). UV-visible absolute absorption spectra (270–700 nm) were recorded at room temperature using a NanoVue,4282 single beam spectrophotometer (CE Healthcare, Cardiff, UK) (Appendix A). CO-binding difference absorption spectra (400–500 nm) were recorded using a dual-beam Shimadzu UV-2600i spectrophotometer as follows. The CO gas was bubbled through the protein solution, which was then divided into two cuvettes serving as a blank and standard, respectively. The baseline was taken, and a few crystals of sodium dithionate were added to the sample cuvette (Appendix A). The corresponding spectra of the original protein sample that was used for crystallization were taken for comparison.

## Figures and Tables

**Figure 1 ijms-26-05689-f001:**
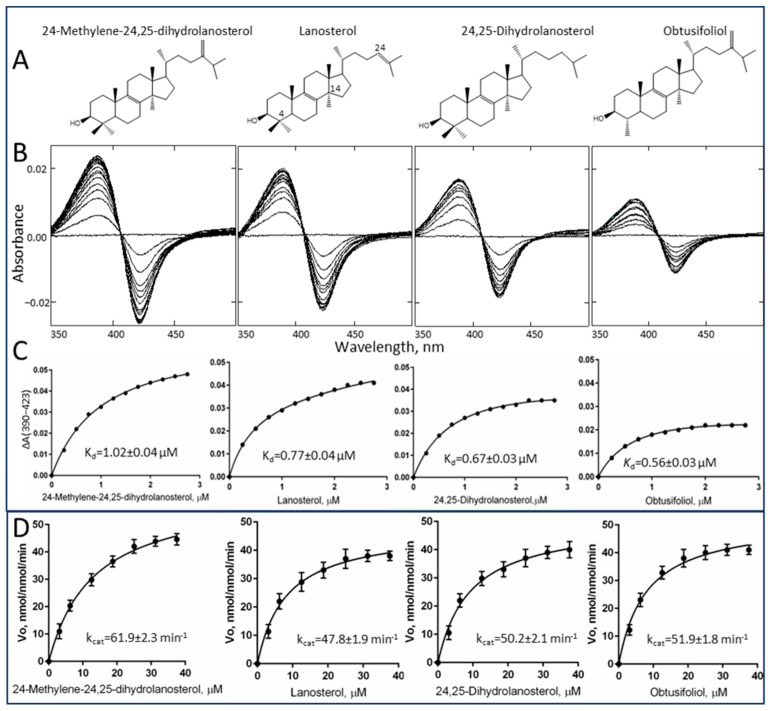
Substrate binding and catalysis by *C. armatus* CYP51. (**A**) Structural formulas of sterol substrates. (**B**) Difference absorbance spectra in response to the sterol addition. Sterol concentration range 0.25−3 µM, titration step 0.25 µM. P450 concentration 2 µM, optical path length 1 cm. (**C**) Titration curves. (**D**) Michaelis–Menten plots at 0.25 µM P450 and 1 µM CPR, 1 min reaction. Sterol concentration range 3.12−37.5 µM. The Michaelis–Menten kinetic experiments were performed in triplicate; the results are presented as mean ± SD. The corresponding calculated binding and steady-state kinetic parameters are given in Table 1.

**Figure 2 ijms-26-05689-f002:**
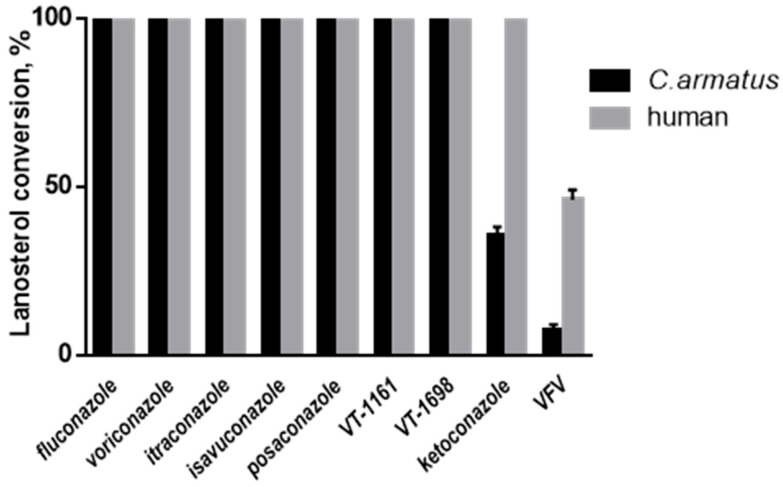
Inhibitory effects of clinical antifungal drugs and an experimental inhibitor VFV on the enzymatic activity of *C. armatus* and human sterol 14α-demethylases. The incubation time was 60 min, the molar ratio enzyme/inhibitor/substrate was 1:50:50, and the P450 concentration was 0.25 µM. The experiments were performed in triplicate; the results are presented as mean ± SD.

**Figure 3 ijms-26-05689-f003:**
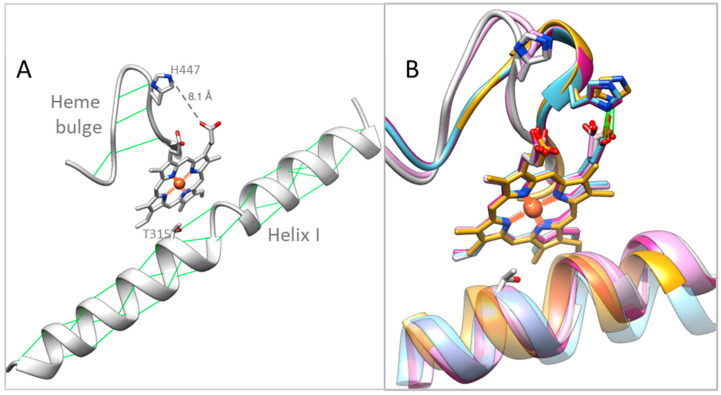
Helix I and the heme bulge in the CYP51 structures. (**A**) Ligand-free *C. armatus* CYP51 [9BAT]. The main-chain hydrogen bonds and the side-chain hydrogen bond donated by the T315 hydroxyl group are depicted as green lines. The distance between imidazole ring nitrogen of H447 and the heme ring D propionate is shown as a gray dashed line and marked. (**B**) Enlarged view of *C. armatus* CYP51 superimposed with ligand-free human [8SBI, plum], ligand-free *Trypanosoma brucei* [3G1Q, gold], substrate-bound [8SS0, blue] and inhibitor-bound [5Q2T, magenta] human CYP51. The conserved His-heme salt bridge is shown in green. The I helix is semitransparent”.

**Figure 4 ijms-26-05689-f004:**
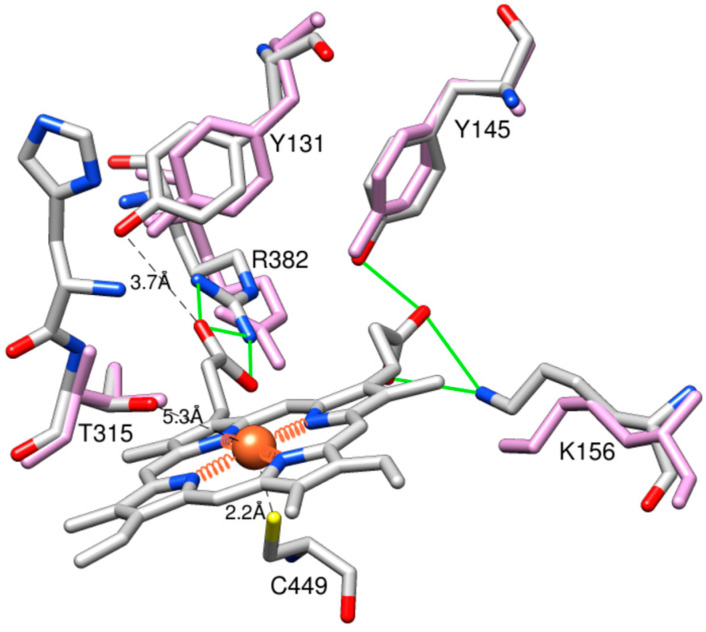
Heme surroundings in the 9BAT and 8SBI structures. The residues are colored by heteroatoms; the *C. armatus* CYP51 carbons are grey. The corresponding residues in human CYP51 are colored in plum.

**Figure 5 ijms-26-05689-f005:**
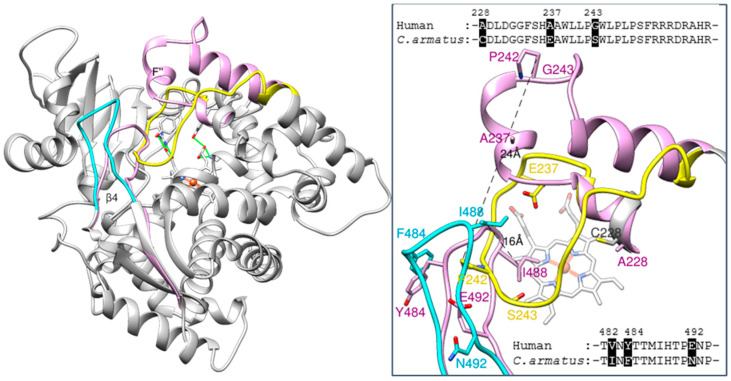
Positions of the FG arm (yellow) and the tip of the β4 hairpin (cyan) in the *C. armatus* and human (plum) CYP51 structures. The rest of the ribbon- of *C. armatus* CYP51 is grey. The heme and supporting residues are displayed as reference points. For clarity, the two-amino acid residue fragment missing in the structure (F234 and S235) was built in MOE (using protein structure preparation). Inset: Enlarged view of the FG arm and β4 hairpin in both structures. The label colors correspond to the color of the ribbons. Distances between the Cβ atoms of P242 (the end of the F’’ helix in 8SBI) as well as between the Cβ atoms of I488 (the tip of the β4 hairpin in both structures) are marked. Sequence alignments of the FG arm and B4 hairpin fragments are shown above and below, respectively. Superimposition of these two structures with selected other human CYP51 structures available in PDB (3JUV, 4UHI and 6UEZ) can be seen in Appendix A. The orientation is about the same.

**Figure 6 ijms-26-05689-f006:**
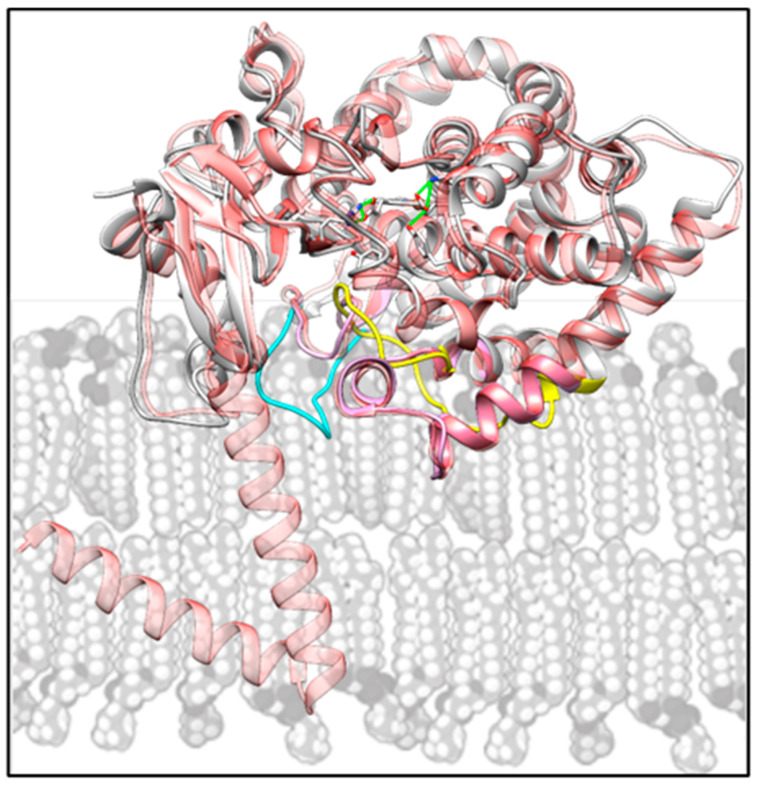
Schematic representation of the CYP51 position within the lipid bilayer of the endoplasmic reticulum (ER) membrane (grey). The *C. armatus* (9BAT) and human (8SBI) structures (colored as in Figure 4) are superimposed with the AlphaFold model of full-length human CYP51 (AF-Q16851-F1, semitransparent red), whose N-terminal helical anchor crosses the membrane.

**Figure 7 ijms-26-05689-f007:**
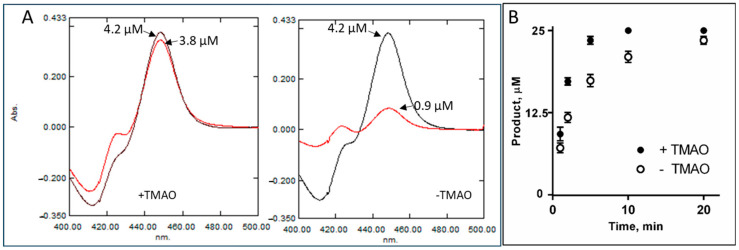
Stabilizing effects of TMAO on *C. armatus* CYP51. (**A**) CO binding spectra of the protein samples in the presence of 10 mM TMAO (left panel) and in its absence (right panel). The initial CO spectra are in black, and the spectra after the overnight storage of the samples at 4 °C in PBS are in red. (**B**) Time course of lanosterol 14α-demethylation, 37 °C, 0.25 µM P450, 1 µM cytochrome P450 reductase, 25 µM lanosterol. The expe were performed in triplicate; the results are presented as mean ± SD.

**Table 1 ijms-26-05689-t001:** Substrate binding and steady state kinetic parameters of *C. armatus* CYP51.

Sterol Substrate	Spin State Transition in the Heme Iron, %	Change in Absorbance, ΔA_max_ per µM P450	K_d_, µM	Binding Efficiency, ΔA_max_/K_d_	k_cat_, min^−1^	K_m_, µM	Specificity Constant, k_cat_/K_m_
Lanosterol	22	24	0.77 ± 0.04	31	47.8 ± 1.9	8.1 ± 1.1	5.9
24,25-Dihydrolanosterol	18	20	0.67 ± 0.03	30	50.2 ± 2.1	9.1 ± 1.2	5.5
24-Methylene-24,25-dihydrolanosterol	27	30	1.02 ± 0.04	29	61.9 ± 2.3	13.0 ± 0.9	4.8
Obtusifoliol	13	14	0.56 ± 0.03	25	51.9 ± 1.8	7.9 ± 0.6	6.6

**Table 2 ijms-26-05689-t002:** Crystallographic data collection and refinement statistics.

Organism [PDB ID]	*C. armatus* [9BAT]	Human [8SBI]
*Data collection* Beamline	ID-23-2, ESRF	21-ID-F, LS-cat
Wavelength, Å	0.9677	0.97872
Space group	P12_1_1	P2_1_2_1_2
Cell dimensions		
a, b, c, Å	69.596, 63.135, 104.860	143.810, 55.640, 103.070
α, β, γ, °	90.0, 97.59, 90.0	90.00, 90.00, 90.00
Molecules per asymmetric unit	2	2
Resolution (upper shell), Å	29.09–2.90 (2.96–2.90)	49.01–2.73 (2.80–2.73)
Solvent content, %	45	51
R_merge_ (upper shell)	0.075 (0.726)	0.12(0.87)
CC (1/2) (upper shell)	1.000 (0.786)	0.996 (0.627)
I/σ(I) (upper shell)	27 (2.5)	9.1(1.3)
Completeness (upper shell), %	96.9 (98.1)	98.2 (96.2)
Redundancy (upper shell)	5.9 (5.4)	4.0 (3.9)
*Refinement*		
No. of unique reflections	18,679	22,239
R_work_/R_free_	0.232/0.248	0.230/0.246
R.m.s deviations		
Bond lengths, Å	0.004	0.007
Bond angles, °	1.06	1.1
Ramachandran plot		
Favourable/allowed, %	93.5/99.8	97/100
Outliers, %	0.2	0
Average B factor, Å^2^	98.0	51.2
*Model*		
No. of atoms	7173	7299
No. of residues per molecule		
Protein (B factor, Å^2^, A/B/)	445 (91.8/106.2)	445 (51/55.5)
Heme (B factor, Å^2^) Water (B factor, Å^2^)	1 (66.7/67.5) 22 (52.0)	1 (50/52.0) 93 (44)

## Data Availability

The authors declare that the data supporting the findings of this study are available within the article and its Appendix A. The structural factors and coordinates of human and *C. armatus* CYP51 are available at the Protein Data Bank, under the accession codes 8SBI and 9BAT, respectively. The cDNA sequence of *C. armatus Cyp51* is available at GenBank under the accession code PV425002. All other data that support the results of this study are available from the corresponding author upon request.

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
