# Peer review of "Unique Structural Features Relate to Evolutionary Adaptation of Cytochrome P450 in the Abyssal Zone"

_ijms, 2025, doi:10.3390/ijms26125689_

Round 1

Reviewer 1 Report

Comments and Suggestions for Authors

In this study, Tatiana Y. Hargrove et al reported a crystal structure of cytochrome P450 from the abyssal fish Coryphaenoides armatus, in which its catalytic domain has more than 80% sequence identity with that of humans and other fish species. While C. armatus CYP51 has specific or more dynamic FG arm and β4 hairpin compared to other solved homologue structures. They found that all the new features of the resolved C. armatus CYP51 structure were related to the low-temperature and high-pressure deep-sea environment. For example, the degree of entrance closure of the active site of CYP51 in fish is lower than that in humans, which could explain the low concentration of substrates in the deep-sea environment; The volume of the cavity in CYP51 structure is reduced by 18%, which may resist high-pressure deformation by reducing molecular flexibility; the increased positive charge surface density in CYP51 structure may be caused by the high salt density in the deep-sea. Their findings suggest an intriguing link between protein structure evolution and the environment. More important, the unique feature they solved in the crystal structure of C. armatus CYP51 has not been predicted by alpha-fold server, implying the necessity of experimental verification. Although there needs evidence for functional relevance in the future, like how the deep-sea fish specific lipid interacts with CYP51 to maintain its activity, and more mutants assay to show the clearly link between the active site structural changes and its enhanced function. But I think this research is interesting and will be of interest to all other readers. I also recommend the author to add some sentences in the discussion to encourage people to perform more studies of CYP51 by using the novel cryo-EM or develop more specific enzyme activity assay method.

In terms of this manuscript, I only have some minor comments:

1) In line 47, the texts representing three bacteria are missing.

2) In Figure 1, why are the kd and Kcat values ​​not presented as mean ± SD as in Table 1 ?

3) In Figure 4, the text size of the distance can be increased.

Reviewer 2 Report

Comments and Suggestions for Authors

The work of Lepesheva et al. continues their cycle of studies on the structure and conformational dynamics of CYP51 sterol 14α-demethylating enzymes.  The authors report the expression, purification, functional characterization, and  X-ray structure of CYP51 from moderately piezophilic abyssal fish Coryphaenoides armatus.  Besides adding the first non-human enzyme to the list of CYP51 proteins of vertebrates with a resolved X-ray structure, this work also represents the first structural study of a P450 protein from a piezophile, thus providing insight into the mechanisms of piezophilic adaptation in proteins.  The study is carried out at a state-of-the-art methodological level and is well-written.  

I have only a few comments and concerns:

  1. Striking structural differences between C. armatus protein and other known CYP51 structures in the position of the β4 hairpin and conformation of the FG arm represent the study's most important finding. However, authors may miss an essential aspect when discussing these intriguing features.  In fact, they studied at ambient pressure a protein whose structure is designed to fold and function at high pressure.  Increased pressure dramatically changes the protein conformation.  To fold and function at their physiological conditions, the piezophilic proteins must have conformational landscapes at high pressure similar to those of their non-piezophilic homologs at ambient pressure.  Placing a piezophilic protein at ambient pressure displaces its native conformational equilibrium towards low-populated states, which may be transient in catalysis (see http://dx.doi.org/10.4172/2157-7544.1000e110 for a discussion).  Thus, it would be logical to hypothesize that the peculiar conformation of CYP51 from C. armatus observed by the authors does not represent the predominant conformation of the substrate-free enzyme observed at the pressure of habitation of the fish.  Instead, it is a transient, low-populated conformation common to all CYP51 enzymes, which becomes more populated due to the placement of the high-pressure-adapted protein at ambient pressure.  In my opinion, discussing this possibility would strengthen the manuscript considerably.
  2. The fragment of discussion about the effect of TMAO raises several concerns:
    1. Why the experimental data on the effect of TMAO were placed in Discussion? Their proper place is in Results.
    2. The data on the effect of TMAO on the stability of CYP51 has no relevance to the physiological function of TMAO in piezophiles and its effect on CYP51 in vivo. TMAO is an osmolyte that "sucks" water from proteins, thus protecting them from pressure-induced hydration.  To exhibit this effect, TMAO should be present at high concentrations that change osmolality significantly.  According to the paper from the lab of Paul Yancey (Gillett et al, 1997; https://doi.org/10.1002/(SICI)1097-010X(19971101)279:4<386::AID-JEZ8>3.0.CO;2-K), while the concentration of TMAO in the muscles of shallow-water fish typically occurs at 20-70 mM, its concentration in C. armatus is as high as 173 mM.  In this view, the concentration of 10 µM used by the authors has no relevance to its physiological function.  Indeed, this concentration's apparent "allosteric" effect is curious but has nothing in common with the TMAO effect in vivo.  In this view, I would suggest removing these data from the manuscript.  
    3. However, if the authors prefer to keep it, they will need to do something with Fig. 5B that looks confusing. In this figure, which illustrates the effect of TMAO on the kinetics of substrate metabolism, the Y-axis is labeled "Substrate, %." Shouldn't it be "Product, %"?  Is it the percentage of the accumulated product from the total concentration of the substrate?  Or do the authors mean the percentage of accumulated product from the maximal level of its formation?  It must be specified.  But why is this graph presented with a relative Y-axis at all?  With a percentage (of whatever) on the Y-axis, the actual effect of TMAO on the reaction rate is hidden.
